# Effect of Embryo Vitrification on the Steroid Biosynthesis of Liver Tissue in Rabbit Offspring

**DOI:** 10.3390/ijms21228642

**Published:** 2020-11-16

**Authors:** Francisco Marco-Jiménez, Ximo Garcia-Dominguez, Marta Domínguez-Martínez, María Pilar Viudes-de-Castro, Gianfranco Diretto, David S. Peñaranda, José S. Vicente

**Affiliations:** 1Laboratory of Biotechnology of Reproduction, Institute for Animal Science and Technology (ICTA), Universitat Politècnica de València, 46022 Valencia, Spain; fmarco@dca.upv.es (F.M.-J.); ximo.garciadominguez@gmail.com (X.G.-D.); marta.dominguez@mdc-berlin.de (M.D.-M.); dasncpea@upvnet.upv.es (D.S.P.); 2Centro de Investigación y Tecnología Animal, Instituto Valenciano de Investigaciones Agrarias, 12100 Segorbe, Spain; viudes_mar@gva.es; 3Italian National Agency for New Technologies, Energy and Sustainable Development (ENEA), Casaccia Research Centre, 00123 Rome, Italy; gianfranco.diretto@enea.it

**Keywords:** embryo, assisted reproductive technologies, cryopreservation, stress, metabolome, cholesterol, IGF-I, steroid biosynthesis, RT-qPCR

## Abstract

Preimplantation embryo manipulations during standard assisted reproductive technologies (ART) have significant repercussions on offspring. However, few studies to date have investigated the potential long-term outcomes associated with the vitrification procedure. Here, we performed an experiment to unravel the particular effects related to stress induced by embryo transfer and vitrification techniques on offspring phenotype from the foetal period through to prepuberal age, using a rabbit model. In addition, the focus was extended to the liver function at prepuberal age. We showed that, compared to naturally conceived animals (NC), offspring derived after embryo exposure to the transfer procedure (FT) or cryopreservation-transfer procedure (VT) exhibited variation in growth and body weight from foetal life to prepuberal age. Strikingly, we found a nonlinear relationship between FT and VT stressors, most of which were already present in the FT animals. Furthermore, we displayed evidence of variation in liver function at prepuberal age, most of which occurred in both FT and VT animals. The present major novel finding includes a significant alteration of the steroid biosynthesis profile. In summary, here we provide that embryonic manipulation during the vitrification process is linked with embryo phenotypic adaptation detected from foetal life to prepuberal age and suggests that this phenotypic variation may be associated, to a great extent, with the effect of embryo transfer.

## 1. Introduction

There is currently evidence that cellular manipulations during standard assisted reproductive technologies (ART) have inherent consequences for embryo development [1,2,3]. Among reproductive technologies, cryopreservation produces the most hostile conditions for embryos, and consequently, an increased number of studies have demonstrated both a direct effect on embryos and long-term effects on gene expression, proteome and metabolome on adult animals, providing a modified foetal and postnatal phenotype [4,5,6,7,8,9,10,11]. The early stage of mammalian development from fertilisation to implantation is a period in which global changes in the epigenetic scenery take place [1,12,13,14,15]. In ART, all of those global changes take place under in vitro conditions when the embryos develop in culture conditions that attempt to mimic the uterine environment. Current systems of in vitro culture do not adequately represent what happens in vivo, and consequently, developed embryo notably differ from the in vivo counterparts [16,17]. There are several lines of evidence suggesting that, depending on the nature of the embryo manipulation, there will be changes affecting the epigenetic reprogramming, having an impact on the individual [12,14,15,18]. Even though the use of ART has been noted as safe, increasing findings make it necessary to assess potential risks in adulthood to improve the safety of the reproductive techniques [1,3,19,20,21,22]. Cryopreservation, a routine technique in both livestock propagation and application in other fields such as human medicine [23], is a technique with significant cellular manipulation associated with imprint disruptions [12]. Cryopreservation is a technique in which embryos are exposed to toxic chemical factors and slight nonphysiological temperatures [24]. This technique preservation technique has become increasingly popular (+13.6%), settling as the second most commonly used in fertility treatments [25].

Few studies to date have investigated the potential long-term outcomes associated with the cryopreservation procedure. We previously developed a cohort of studies using a rabbit model under strictly regulated experimental conditions, linking cryopreservation with short- and long-term results. We assessed the gene expression in preimplantation embryos [6,7,8,26], the foetal and placental weights [8], the foetal placental transcriptome and proteome [7,8], the skewed sex ratio [27], the offspring’s growth pattern and adult body weight [11,27,28] and the reproductive traits derived from females [11,29] and males [11]. More recently, we demonstrated the transgenerational effects following a vitrified embryo transfer procedure [30]. Taken together, these data showed transgenerational phenotypic changes in response to vitrification. Moreover, there is a growing body of studies evidencing that ART manipulations are associated with phenotypic outcomes in human [20,21,31,32,33,34,35] and animals species [19,20,22,31,36,37,38,39,40,41]. However, it has been challenging to connect the changes generated by different ART techniques when the endpoint is the offspring [12]. To fill this gap, the current study evaluated the particular effects related to the embryo transfer and vitrification technique on offspring from the foetal period through to prepuberal age using a rabbit model. The liver function at prepuberal age was also assessed to detect possible alterations induced by the vitrification procedure.

## 2. Results

### 2.1. Foetal and Postnatal Growth Performance, Body Weight and Organ Phenotype Study

The CRL foetus from 11 to 22 days of gestation registered significant differences. Both FT and VT groups showed a lower foetal length and foetal placental area, disappearing afterwards (Figure 1a,b, respectively).

At parturition, differences were found between survival rates of FT and VT embryos (64.1% (124/194) vs. 50.4% (127/252) for the FT vs. VT group, respectively, *p* ≤ 0.05). The average litter size was 9.7 ± 0.37 for the NC group (79 offspring), 7.7 ± 0.38 for the FT group (124 offspring) and 7.1 ± 0.45 for the VT group (127 offspring). VT progenies showed higher birth weight compared to the FT and NC groups (59.5 ± 0.87 vs. 56.0 ± 0.794 and 53.0 ± 1.24 g, for the VT vs. FT and NC groups, respectively, *p* ≤ 0.05. Figure 2a). At four weeks old, body weight VT and FT progenies showed a higher birth weight compared to the NC group (639.8 ± 12.93 and 642.2 ± 11.57 vs. 601.00 ± 18.52 g, for the VT vs. FT and NC groups, respectively, *p* ≤ 0.05. Figure 2a). At nine weeks old, body weight VT and FT progenies showed a higher birth weight compared to the NC group (1908.7 ± 23.96 and 1846 ± 22.88 vs. 1795.58 ± 30.06 g, for the VT vs. the FT and NC groups, respectively, *p* ≤ 0.05. Figure 2a). However, VT progenies showed a significantly higher body weight at nine weeks old compared to the NC and FT groups. On the other hand, VT animals showed the highest liver weight, followed by the FT and NC groups (66.9 ± 1.46 vs. 60.9 ± 1.22 and 56.8 ± 1.30 g, for the VT vs. FT and NC groups, respectively, *p* ≤ 0.05. Figure 2b). FT showed a higher lung weight (17.3 ± 0.52 vs. 14.5 ± 0.62 and 14.8 ± 0.54 g, for the FT vs. VT and NC groups, respectively, *p* ≤ 0.05), but with respect to the adrenal gland’s weight, FT and VT groups, which maintained higher values (0.3 ± 0.014 vs. 0.3 ± 0.017 and 0.2 ± 0.015 g, for FT vs. VT and NC group, respectively, *p* ≤ 0.05. Figure 2b). No significant differences were observed for the rest of the organs.

### 2.2. Hepatic Functionality Assessment

Attending to the serum biochemical marker results, lower levels of cholesterol were exhibited by the VT animals (56.9 ± 6.3 vs. 66.4 ± 6.6 and 80.2 ± 6.34 mg/dL, for the FT vs. VT and NC groups, respectively, *p* < 0.05. Figure 3). Lower levels of IGF-I were shown by the VT and FT animals compared to NC (293 ± 13.3 and 279 ± 12.9 vs. 334 ± 13.6 mg/dL, for the VT and FT vs. NC groups, respectively, *p* < 0.05. Figure 3). No significant differences were observed for the rest of the parameters. Levels ranged between the normal values.

### 2.3. Comparative Study of Liver mRNA Expression

In two of the six candidate genes tested (*APOA4* and *CLN6*), reduced mRNA expression was detected. Correctly, significant differences were observed in mRNA expression of *APOA4* between VT and NC and CLN6 between FT and NC groups (Figure 4). No significant differences were found for the rest of the genes.

### 2.4. Comparative Study of Liver mRNA Expression

The score plots of PCA and HM of untargeted liver metabolites showed clear discrimination between FT, VT and NC samples (Figure 5a,b). Statistically significant differences in levels of 661 metabolites were noted between FT and NC, while 557 were noted between VT and NC (Figure 5c). Examining the relationships between FT and VT procedures, a total of 96 untargeted metabolites were significantly altered (Figure 5c). Remarkably, only 18 metabolites were induced by the vitrification technique (Figure 5c). Moreover, after targeted metabolomic analysis, a total of 27 relatively quantified significant metabolites involved in cholesterol synthesis and catabolism were obtained.

The complete metabolite dataset, its FC and DAMs are reported in Table 1. Most of the studied metabolites were downaccumulated (18 metabolites) in FT and VT samples. 3-alpha,7-alpha,12-alpha,26-Tetrahydroxy-5beta-cholestane metabolite was specifically downaccumulated only in VT samples (Table 1). At the same time, a general upaccumulation in lanosterol, S-squalene 2,3-epoxide and 20-alpha,22-beta-dihydroxycholesterol was found in VT and FT livers over NC (Table 1). Among them, 19 metabolites were detected in steroid biosynthesis (Figure 6).

## 3. Discussion

The present study describes for the first time that embryos exhibit a nonlinear synergistic relation between cumulative stressors after ART from foetal life to prepuberal age. More in detail, we show that, compared to NC, progeny generated after embryo exposure to the cryopreservation procedure (VT) displayed variation in growth and body weight, and further in liver function, with no direct evidence that it impacts animal health. Our data constitute the first report of metabolomics alterations in offspring’s liver biopsies born after ART. The present major novel findings include the significant alteration of the liver metabolome and steroid biosynthesis profiles in animals born from transfer procedure or vitrified embryos procedure compared to naturally conceived animals.

Cohort studies from humans and animal models evidence that ART-conceived offspring have detected differences in body weight [19]. Therefore, our study adds further evidence for the short- and long-term effects of ART on phenotype. Bodyweight is used as phenotypic criteria for growth and is considered to reflect proliferative events during organogenesis and development [42]. In connection with this, our results supported the notion that both fresh transfer and vitrification processes are stressors associated with changes in growth pattern and body weight from foetal life to prepuberal age [11,28,43]. The effect of ART manipulation on foetal development restriction during early or mid-gestation has been well described [19]. In fact, the significantly lower growth was already shown for the VT foetus compared to FT [7,8]. In the present study, we demonstrated that both FT and VT placenta and foetus are reduced in size compared with NC until late-gestation [44]. It is worth mentioning that at late-gestation, no differences in foetal and placental weights at 24 days of gestation between both ART manipulation were observed, although transcriptomic and proteomic changes were detected [7,8]. Our study revealed that ART-conceived animals have an increased incidence of higher birth weight and growth kinetics until prepubertal age, suggesting that both stressors were involved in the bodyweight variation. As vitrification is accompanied by the embryo transfer procedure to generate offspring, we analysed the impacts of individual stressors due to negative synergies that can exist when more than one stressor is present [19,45]. We detected that fresh transfer embryo manipulation is linked to both growth pattern and body weight, in line with our recent results [11]. Our study analysed two stressors that could act synergistically, resulting in cumulative effects [19]. Using a metabolome approach, we found evidence of a slight difference in “global” metabolome between fresh transfer and vitrification offspring groups at prepuberal age. Thus, we highlighted that only 18 differentially accumulated nonpolar metabolites were induced by the vitrification technique. These findings suggest a nonlinear relationship between both stressors.

The primary goal of ART is to achieve successful pregnancies to term; therefore, the transfer procedure will always be mandatory. Although embryonic manipulation during the recovery (flushing, washing and scoring) and transfer processes can be considered less invasive, our results demonstrated that this stressor generates notable outcomes from foetal life to prepuberal age. In fact, phenotypic variation during the vitrification process may be associated with, to a large extent, the effect of the embryo transfer technique. Therefore, it can be a good starting point to understand this form of stress and perhaps be able to manipulate embryos in different ways to partially alleviate potential underlying molecular and cellular variations [1].

The molecular mechanisms that align early embryo manipulation with the growth performance have up to now remained poorly understood. In part, the results presented herein could help to fill this gap. In more detail, we studied the liver tissue as a means of understanding the nature of growth differences. As a first analysis, we identified differences in liver and adrenal gland weights in the offspring derived from both embryonic manipulations, both glands with essential metabolic and endocrine activity related to the use of cholesterol and, in general, lipid metabolism. We accurately appreciated a cumulative effect in VT animals comparing FT and NC, even though the liver weight was corrected by body weight for the analysis, consistent with our previous results [28]. Because the liver is the main metabolic organ in the body, serving as a significant hormonal secretory gland and functioning to maintain hormone balance and homeostasis [46], we investigated the association between liver markers and embryonic manipulation. Thus, we found that the serum *IGF-I* and cholesterol levels in FT and VT slightly declined when compared with the NC animals. It is important to note that, although we found evidence for a specific effect of ART conception on growth, serum *IGF-I* and cholesterol and liver metabolome, normal values for a conventional blood test for liver function were obtained in all experimental groups [11]. It is well established that the liver is the major contributor to circulating *IGF-I* and cholesterol [46,47]. The serum level of *IGF-I* plays an essential role in postnatal growth [48], although it is not necessary for postnatal growth and development [47]. Until now, little data exist about the effects of ART manipulation on *IGF* levels in offspring. In a case–control study, IVF children were taller with higher *IGF* levels compared to NC [49]. In contrast, ICSI-conceived children had slightly lower serum *IGF-I* than NC children at three years old. So far, results arising in rabbits have been consistent with those derived from human embryos. It is well established that free *IGF*-I levels increased during childhood, with the highest mean values during puberty and a subsequent decline [50]. We hypothesise that slightly lower *IGF* levels in FT and VT animals would be explained by an accelerated growth compared to NC, coinciding with a higher body weight observed at nine weeks old. Several studies have reported a relationship between IVF-ICSI children and lower total cholesterol [35,51]. In this study, low cholesterol levels might be linked to a downexpression of *APOA4* expression in the hepatic tissue of both VT and FT animals. *APOA4* is a key molecule that mediates cholesterol efflux [52,53,54]. Decreased expression of the *APOA4* has been associated with reduced levels of plasma triglyceride and total cholesterol [55]. This result suggests that the lower expression of *APOA* can contribute significantly to the reduction of free cholesterol in the liver and therefore cause its decline in the blood. This hypothesis was confirmed when the analysis was expanded to detect cholesterol in liver tissue. In both FT and VT animals, indeed, hepatic cholesterol was downaccumulated compared to NC. To better elucidate alterations in cholesterol metabolism, we performed a targeted metabolomics analysis of compounds taking place in biosynthesis and metabolism. In this way, interestingly, 27 differentially accumulated metabolites in the cholesterol pathway identified in offspring underwent VT and FT procedures. Furthermore, 19 differentially accumulated metabolites were involved in steroid biosynthesis (Figure 5). In combination, these data suggest that stress inside the normal range during ART manipulation induces an adaptive response changing that influence developmental trajectory in adulthood [2,3,20]. In 2000, Lee et al. proposed the Goldilocks’ principle, which could explain what seems to be happening with embryo manipulation. Early embryos display a remarkable flexible homeostatic mechanism to allow for the capacity to upregulate or downregulate metabolism in response to stress within the optimum [56]. Thus, the embryo homeostatic mechanism up or down within the optimum from which the embryo can overcome, while extreme perturbation surpasses the optimum, which shifts metabolism irreversibly. Although the process explaining differential susceptibility to cryopreservation in embryos is not understood [57], our findings suggest the activation of molecular mechanisms and cellular events involved in an adaptive response to the lethal effects of vitrification. In recent years, epigenetic alterations have been invoked as a mechanistic candidate that might explain the association between ART and its associated outcomes throughout life [1,12,15,18,58]. At present, the link between epigenetic modifications and long-term effects in terms of phenotype in ART offspring remains unclear.

In conclusion, embryonic manipulation during the vitrification process is linked with embryo phenotypic adaptation detected from foetal life to prepuberal age. The phenotypic variation during the vitrification process may be associated with a large extent the effect of embryo transfer technique. Additional investigations in both animal models and humans are needed to replicate our findings. The possible health implications do not seem relevant, given the lack of consequences in the liver function test.

## 4. Materials and Methods

Unless stated otherwise, all chemicals were purchased from Sigma-Aldrich Corporation (St Louis, MO, USA).

### 4.1. Animals and Ethical Statements

New Zealand rabbits belonging to the Universitat Politècnica de València were used throughout the experiment. The animal study protocol was reviewed and approved (code number 2018/VSC/PEA/0116) by the “Universitat Politècnica de València” ethical committee prior to the initiation of the study. All experiments were performed in accordance with guidelines and regulations set forth in Directive 2010/63/EU EEC. Animal experiments were conducted at an accredited animal care facility (Code: ES462500001091).

### 4.2. Experimental Design

Figure 7 illustrates the experimental diagram. A total of 20 naturally conceived females were used to compose the control group (NC group, 8) and 12 donors to produce embryos at Day 3 of development, which were pooled and divided into the fresh transfer group (FT) and vitrified-thawed transfer group (VT). A total of 446 embryos (194 fresh and 252 vitrified embryos) were transferred into 34 foster mothers (16 FT and 18 VT with an average body weight of 3736.0 ± 305.6 g and 3668.4 ± 336.7 g for FT and VT, respectively). From 11 to 27 days of gestation, foetal growth and placental area were examined by ultrasonography in 8 females of each group. On their birthdays, offspring were weighed, sexed and microchipped. Body weight was compared at 4 and 9 weeks of age, which coincides with the weaning and prepubertal age, respectively. At 9 weeks of age, biochemical analyses were addressed to evaluate hepatic functionality. Finally, after the ultimate bodyweight was taken, random animals of each sex (52 FT, 58 VT and 42 NC) were euthanised, and the wet liver, lungs, heart, kidneys, adrenal glands and spleen were recorded. In rabbits, body weight at birth, weaning (4 weeks) and adulthood (20 weeks) are not affected by sex [11,27,59]. In rabbits, sexual dimorphism appears after they reach puberty at approximately 13–14 weeks of age [60]. Considering this fact, no sex effect was observed.

### 4.3. Embryo Collection Procedure

Firstly, 12 donor females were superovulated (average body weight of 3635.3 ± 307.3 g), using 3 µg of corifollitropin alpha [61], and inseminated after three days with semen from 10 males (average body weight of 3675.0 ± 141.7 g) [62]. Ovulation was induced by an intramuscular injection of 1 µg of buserelin acetate (Hoechst Marion Roussel, Madrid, Spain). Three days after insemination, late morulae–early blastocyst embryos were recovered by perfusion of each oviduct and uterine horn with 10 mL of prewarmed Dulbecco’s PBS supplemented with 0.2% of BSA. The embryos with the highest morphological quality (based on homogenous blastomeres, intact mucin coat and zona pellucid) were distributed in pools of 12–14 embryos for the fresh transfer group (described below) or for the vitrified-thawed transfer group.

### 4.4. Embryo Vitrification

Embryos were vitrified and thawed according to a highly efficient protocol developed previously to cryopreserve rabbit embryos by vitrification [57,63]. This protocol allows the survival of >80% of the thawed embryos, and we have generated thousands of descendants in our laboratory since its implementation [57]. Briefly, vitrification was achieved in two steps at room temperature (around 22 °C). In the first step, embryos were placed for 2 min in a solution consisting of 10% (*v/v*) dimethyl sulphoxide (DMSO) and 10% (*v/v*) ethylene glycol (EG). In the second step, embryos were suspended for 1 min in a solution of 20% DMSO and 20% EG. Then, embryos were then loaded into 0.125 mL of the French mini straws (IMV Technologies, L’Aigle, France) device and directly plunged into liquid nitrogen to achieve vitrification. Warming was performed by horizontally placing the straw 10 cm from liquid nitrogen for 20–30 s, and when the crystallisation process began, the straws were immersed in a water bath at 20 °C for 10–15 s. The vitrification medium was removed while loading the embryos into a solution containing DPBS and 0.33 M sucrose for 5 min, followed by one bath in a solution of DPBS for another 5 min. After thawing, embryos were successfully transferred into the foster mothers by laparoscopy, following the protocol described below. The vitrification procedure is described in detail in our recent report [64].

### 4.5. Embryo Transfer Procedure

Both fresh and vitrified-warmed embryos were laparoscopically transferred into the oviduct of asynchronous foster mothers, following the protocol described by Besenfelder and Brem [65]. Only receptive females (determined by the vulva colour) were induced to ovulate by injection of 1 µg of buserelin acetate (Hoechst Marion Roussel S.A., Madrid, Spain) 60 h before transfer [62]. Briefly, foster mothers were anaesthetised with xylazine (5 mg/kg; Rompun; Bayern AG, Leverkusen, Germany) intramuscularly and ketamine hydrochloride (35 mg/kg; Imalgene 1000; Merial S.A, Lyon, France) intravenously, and placed in Trendelenburg’s position. Then, embryos were loaded in a 16G epidural catheter, which was inserted through a 17G epidural needle into the inguinal region. Finally, the process was monitored by single-port laparoscopy. The catheter was introduced in the oviduct through the infundibulum to release the embryos. Transfer procedure was described in detail in our recent report [64]. Embryo transfer was carried out in 4 batches.

### 4.6. Foetal Growth Study

Both foetal growth and placental area were examined at Days 11, 13, 15, 18, 20, 22, 25 and 27, by a portable colour doppler ultrasound device (Esaote, Spain) with 7.5-MHz linear probe (4–12 MHz range). The ultrasound examination was performed from right to left with the probe in sagittal orientation and, after the localisation of different foetal areas, 4–6 foetal sac examinations per doe were performed. Foetal growth was determined as the maximum distance from the crown to tail basis with the foetus on a sagittal plane (crown-rump length (CRL) [6]). Foetal placental measurements were determined when the maximal placental surface with the two-lobed foetal was identified [6].

### 4.7. Growth, Body Weight and Organs Weight Study

Survival rates of fresh and vitrified embryos were assessed at birth rate (kits born/total embryos transferred). Bodyweight differences were assessed at birth, the 4th week (weaning) and 9th week (prepubertal age). A total of 152 animals were euthanised, and wet organ weight of liver, lungs, heart, kidneys, adrenal glands and spleen were recorded. Moreover, liver samples were generated by performing liver biopsies randomly from the same individual organ. The tissue was immediately washed with a phosphate-buffered saline solution to remove blood remnants. The sample was directly flash-frozen in liquid nitrogen and stored at −80 °C for the molecular and metabolomic study.

### 4.8. Hepatic Functionality Assessment

At nine weeks old, 24 (12 of each sex) individual blood samples from each experimental group (FT, VT and NC) were obtained from the central ear artery. From each animal, blood was dispensed into a serum-separator tube (Deltalab S.L., Barcelona, Spain). Biochemical analysis of the serum cholesterol, albumin, total bilirubin and bile acids were performed. Briefly, after blood coagulation, samples were immediately centrifuged at 3000× *g* for 10 min, and serum was stored at −20 °C until analysis. Then, a total of 24 samples, eight samples per group, were analysed, and the plasma concentration of total cholesterol, albumin and total bilirubin levels were analysed by colourimetric enzymatic methods. In parallel, bile acids were estimated by photometry. All the methodologies were performed in an automatic chemistry analyser model Spin 200E (Spinreact, Girona, Spain), following the manufacturer’s instruction. All samples were processed in duplicate. Insulin-like Growth Factor-I (IGF-I) was determined by direct enzyme immunoassay technique following the manufacturer’s instructions (IGF-I Elisa Kit, Diagnostic Systems Laboratories, Inc. Texas, USA, DRG International, Inc. Marburg, Germany). The sensitivity of the tests used was 1.1 ng/mL for IGF-I.

### 4.9. Real-Time Quantitative PCR (RT-qPCR) Analysis

We selected a list of 6 genes as potentially relevant for cholesterol metabolism (*APOA*4; Apolipoprotein A-IV, *ELOVL4*; Fatty Acid Elongase 4, *LIPC*; Lipase C, *CLN6*; Transmembrane ER Protein and *IGF-I*; Insulin-like growth factor I). Total RNA was extracted from 12 animals from each experimental group (FT, VT and NC) using the Dynabeads kit (Life Technologies, Carlsbad, CA, USA), following the manufacturer’s instructions. Then, RT was carried out using SuperScript III reverse transcriptase (Invitrogen, Spain) according to the manufacturer’s instructions. The RT-qPCRs were carried out with an Applied Biosystems 7500 (Applied Biosystems, Spain) as described elsewhere [7]. Each RT-qPCR reaction was carried out from 5 µL of diluted 1:10 cDNA template, 250 nM of forward and reverse specific primers (Appendix A) and 10 µL of PowerSYBR Green PCR Master Mix (Fermentas GMBH, Madrid, Spain) in a final volume of 20 µL. The RT-qPCR protocol included an initial step of 50 °C (2 min), followed by 95 °C (10 min) and 42 cycles of 95 °C (15 s) and 60 °C (30 s). After RT-qPCR, a melting curve analysis was carried out by slowly increasing the temperature from 65 to 95 °C, with continuous recording of changes in fluorescent emission intensity. Serial dilutions of the cDNA pool made from several samples were done to assess RT-qPCR efficiency. A ΔΔCt method adjusted for RT-qPCR efficiency was used, employing the geometric average of H2A histone family member Z (*H2AFZ*) and glyceraldehyde-3-phosphate dehydrogenase (*GAPDH*) as a housekeeping normalisation factor [7]. Relative expression of cDNA pool from various samples was used as the calibrator to normalise all samples within one RT-qPCR run or between several runs.

### 4.10. The Hepatic Metabolomic Approach in Response to ART Stressors

Targeted and untargeted LC-APCI-MS analysis of the hepatic nonpolar metabolome was carried as reported before [30]. Untargeted metabolomics was performed using the SIEVE software (Thermofisher Scientific). Briefly, after chromatogram alignment and retrieval of all the detected frames (e.g., ions), differentially accumulated metabolites (DAMs) were detected by a statistical analysis (one-way ANOVA plus Tukey’s pairwise comparison) using the SPSS software (SPSS Inc., Chicago, Illinois, USA). Principal component analysis (PCA) and Heat-Maps (HM) hierarchical clustering of untargeted metabolomes were performed using the ClustVis online software (https://biit.cs.ut.ee/clustvis/). Targeted metabolite identification was performed by comparing chromatographic and spectral properties with authentic standards (if available) and reference spectra, in-house database and literature data, based on the m/z accurate masses, as reported in the Pubchem database (http://pubchem.ncbi.nlm.nih.gov/) for monoisotopic mass identification, or on the Metabolomics Fiehn Lab Mass Spectrometry Adduct Calculator (http://fiehnlab.ucdavis.edu/staff/kind/Metabolomics/MS-Adduct-Calculator/) in the case of adduction detection. Metabolites were quantified relatively by normalisation on the internal standard (DL-α-tocopherol acetate) amounts. In this study, we analysed six samples (pooled from 4 animals); for each biological replicate, at least one technical replicate was carried out.

### 4.11. Statistical Analysis

A general linear model (GLM) was fitted for the analysis of foetal growth, foetal placental area, body weight, organ weights and hepatic functionality parameters, including the experimental group as a fixed effect and the biological and foster mother as random effects. Prior to running the GLM, data were confirmed to show normal distribution and homogeneity of variances. A pairwise comparison test (Tukey adjustment) was used to assess differences between groups. Litter size was used as a covariate for growth/bodyweight correction after birth until prepuberal age, although it remained nonsignificant from the third week of age. In the case of organ weights, data were corrected using body weight as a covariate. Data of relative mRNA abundance were normalised by a Napierian logarithm transformation and also evaluated using a GLM. Data were expressed as least square means ± standard error of means. Differences of *p* ≤ 0.05 were considered significant. All statistical analyses were performed with SPSS 21.0 software package (SPSS Inc., Chicago, IL, USA).

## Figures and Tables

**Figure 1 ijms-21-08642-f001:**
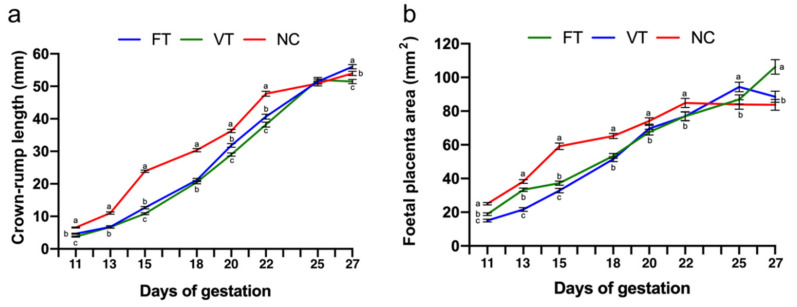
Identification of foetal growth and foetal placental area differences in foetuses born after the embryo manipulation during the fresh transfer procedure (FT) or vitrified-transfer procedure (VT) compared to nonmanipulated (NC) animals. Measures were determined by ultrasonography. (**a**) Crown-rump length of the foetus. (**b**) Foetal placental area. *n* = 40 foetuses in each group. Day of gestation with different superscript letters are significantly different (*p* ≤ 0.05). Data represent the least square means and standard errors of the means.

**Figure 2 ijms-21-08642-f002:**
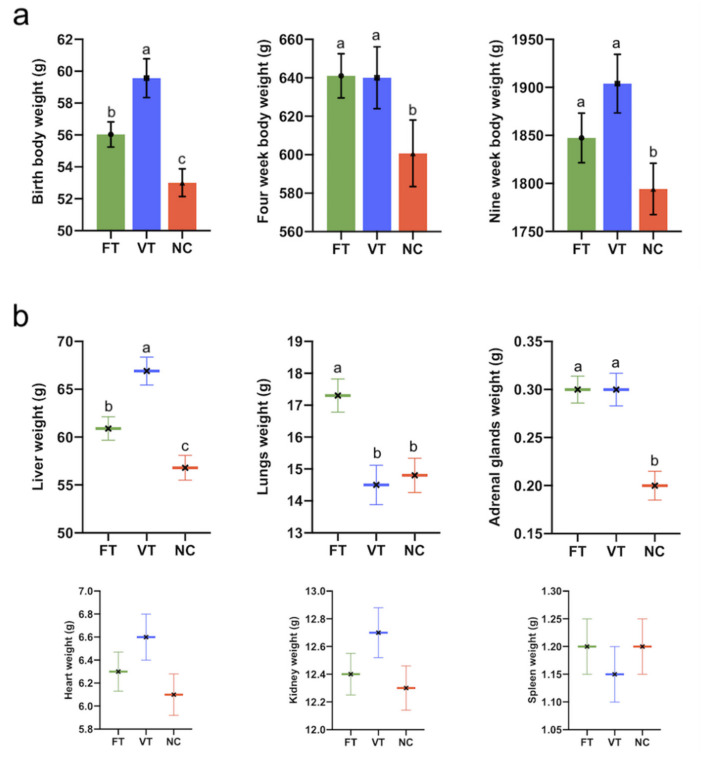
Identification of assisted reproductive technologies (ART)-associated phenotypic and metabolic changes from neonatal to prepuberal age. (**a**) Bars graph shows body weight between offspring born after the embryo manipulation during the transfer procedure (FT, green) or vitrified embryos procedure (VT, blue) compared to nonmanipulated (NC, red) from birth to prepuberal age, showing a significant difference between groups. *n* = 124 FT animals, *n* = 127 VT animals and *n* = 79 NC animals. (**b**) XY plot of the wet organ weight of liver, lungs, heart, kidneys, adrenal glands and spleen, showing a significant difference between groups for liver, lung and adrenal gland. *n* = 52 FT samples, *n* = 58 VT samples and *n* = 42 NC samples. Bars and XY plots with different superscript letters are significantly different (*p* ≤ 0.05). Data represent the least square means and standard errors of the means.

**Figure 3 ijms-21-08642-f003:**
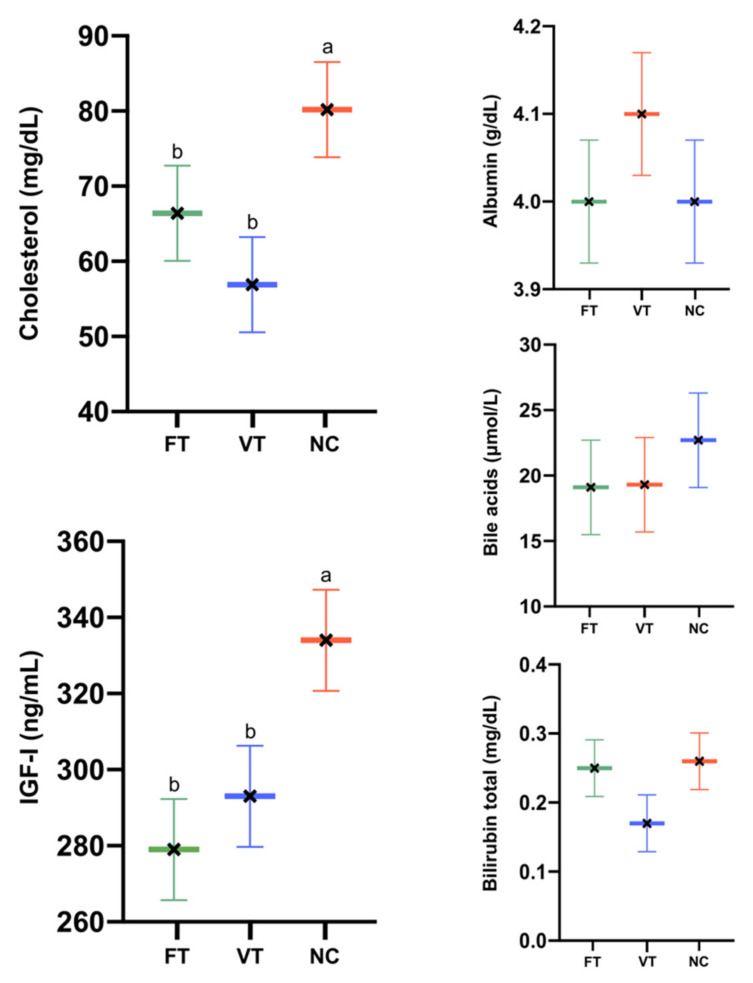
XY plot of serum biochemical marker for hepatic functionality, showing a significant difference between groups for cholesterol and IGF-I. *n* = 20 biologically independent samples for each group. XY plot with different superscript letters are significantly different (*p* ≤ 0.05). Data represent the least square means and standard errors of the means.

**Figure 4 ijms-21-08642-f004:**
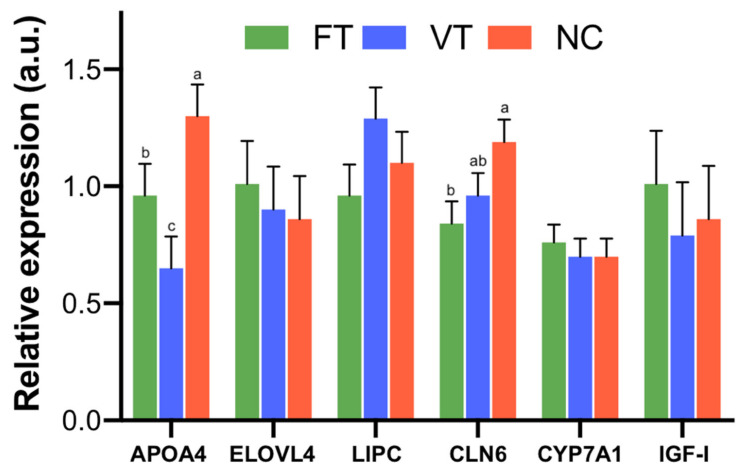
Relative expression of liver tissue was analysed using the six selected genes for RT-qPCR CR normalisation in offspring derived after embryo manipulation during the transfer procedure (FT, green), vitrified-transfer procedure (VT, blue) and nonmanipulated (NC, red). *APOA4*: Apolipoprotein A-IV. *ELOVL4*: Fatty Acid Elongase 4. *LIPC*: Lipase C. *CLN6*: Transmembrane ER Protein. *IGF-I*: Insulin-like growth factor I. Bars with different superscript letters are significantly different (*p* ≤ 0.05). Bars represent the least square means and standard errors of the means of 12 biological replicates.

**Figure 5 ijms-21-08642-f005:**
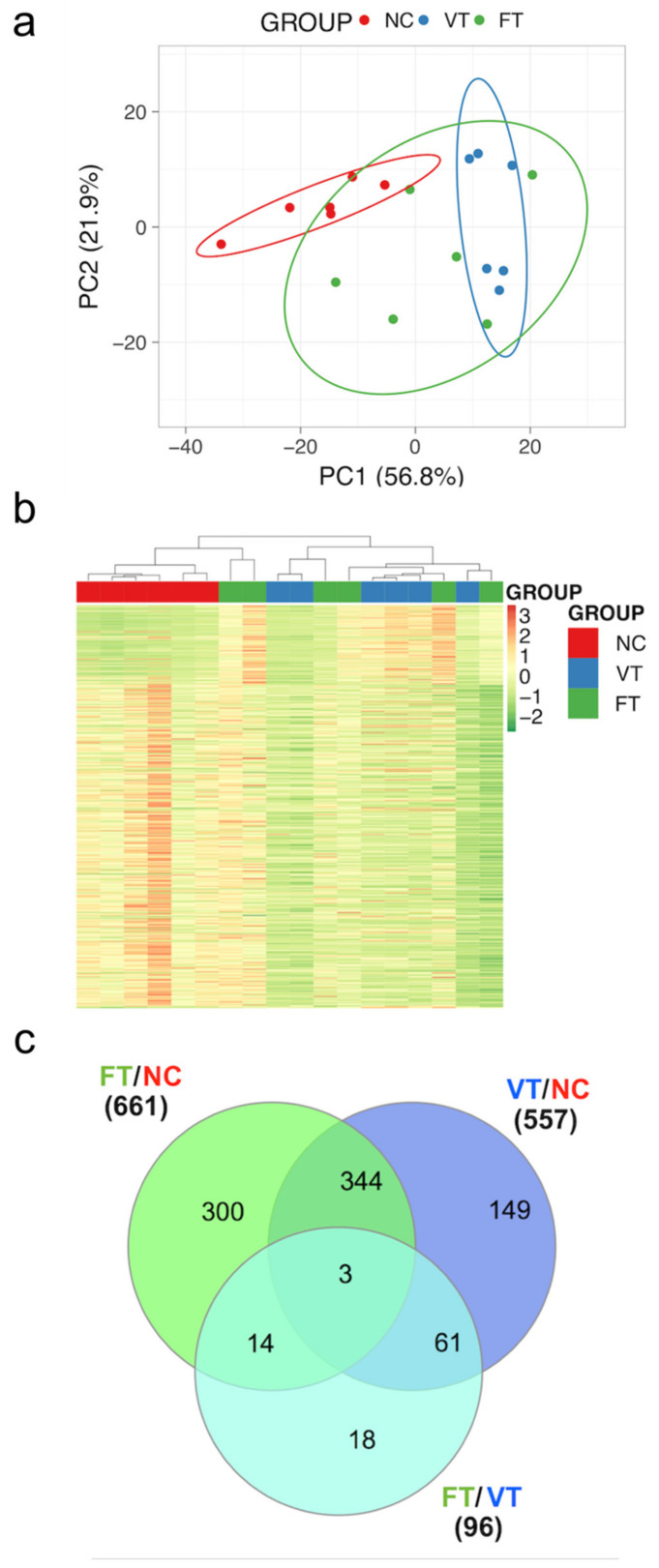
Identification of ART-associated metabolomic and gene expression in prepuberal offspring. (**a**) Principal components analysis and (**b**) Heat-Maps of all differential untargeted metabolites in the liver of offspring born after the embryo manipulation during the transfer procedure (FT, green) or vitrified embryos procedure (VT, blue) compared to nonmanipulated (NC, red). Each is individual plotted. (**c**) Venn diagram showing the overlap among the metabolites altered by the FT vs. NC, VT vs. NC and FT vs. VT for the analysis focused on embryo manipulation.

**Figure 6 ijms-21-08642-f006:**
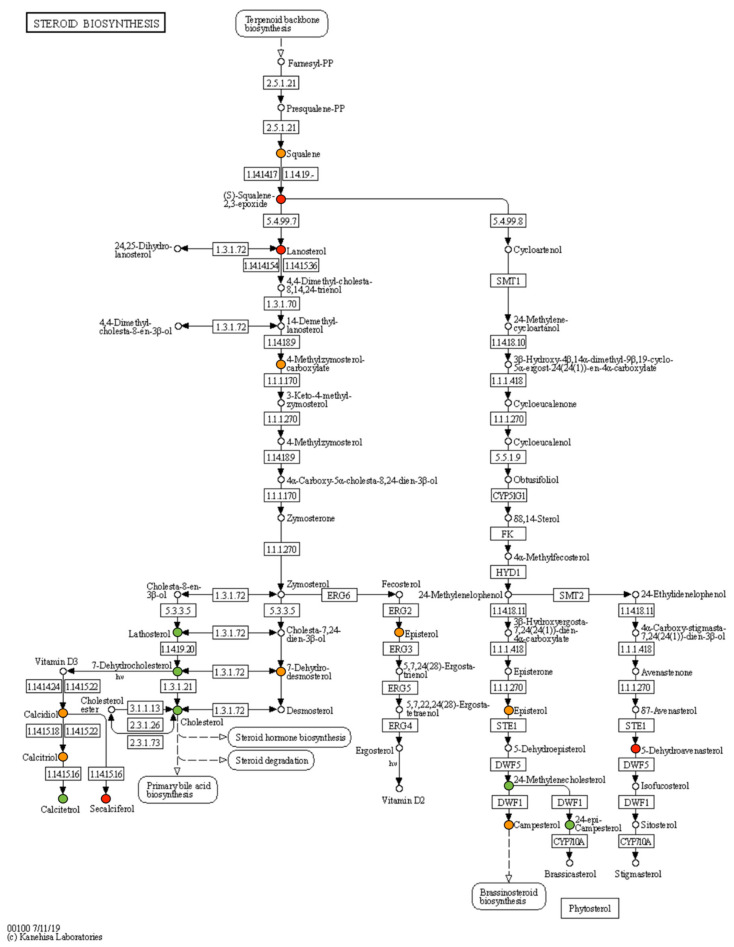
Proposed steroid biosynthesis pathways in Homo sapiens (human). Different colour points were targeted/identified metabolites in prepuberal offspring born after the embryo manipulation during the transfer procedure or vitrified embryos procedure compared to nonmanipulated. Green means downaccumulated metabolite (*p* ≤ 0.05), red means upaccumulated metabolite (*p* ≤ 0.05) and orange means no significant differences.

**Figure 7 ijms-21-08642-f007:**
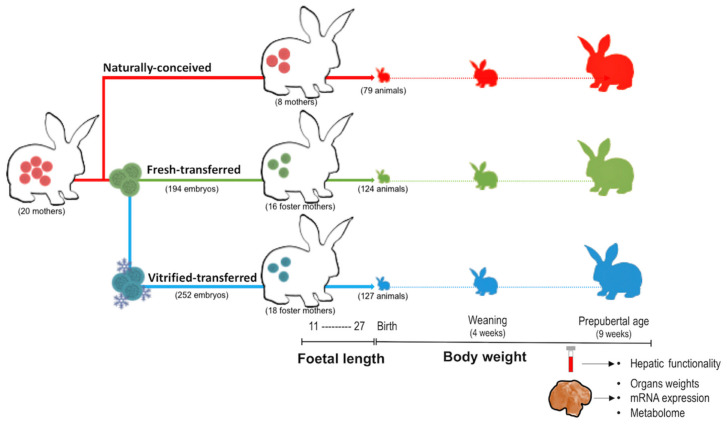
Study design. Summary of the longitudinal phenotypic study of offspring born after the embryo manipulation during the transfer procedure or vitrified embryos procedure compared to nonmanipulated animals from the foetal life to prepuberal age.

**Table 1 ijms-21-08642-t001:** Targeted identification of differentially accumulated metabolites in liver tissue of prepuberal offspring born after embryo manipulation during the transfer procedure or vitrified embryos procedure compared to nonmanipulated in steroid biosynthesis pathway. Red indicates significant differences. FT: fresh-transferred animals. NC: naturally conceived animals. VT: vitrified-thawed transferred animals.

Metabolite	Fresh-Transferred (FT/NC)	Vitrified-Transferred (VT/NC)	Vitrified-Fresh (VT/FT)
	Fold Change	Fold Change	Fold Change
3-alpha,7-alpha,12-alpha,26-Tetrahydroxy-5beta-cholestane	0.11	−0.65	−0.75
3-alpha,7-alpha,26-Trihydroxy-5-beta-cholestane	−1.29	−0.84	0.45
3-alpha,7-alpha-Dihydroxy-5-beta-cholestanate	−3.71	−2.47	1.23
3-alpha,7-alpha-Dihydroxy-5-beta-cholestane	−2.67	−1.87	0.80
4-alpha-Methylzymosterol	−0.77	−0.31	0.46
4-alpha-Methylzymosterol-4-carboxylate	0.57	0.03	−0.54
5-Dehydroavenasterol	0.85	1.45	0.61
7-alpha,24-Dihydroxy-4-cholesten-3-one	−1.20	−0.81	0.38
7-Dehydrocholesterol	−0.90	−0.59	0.31
7-Dehydrodesmosterol	−0.62	−0.30	0.32
14-Demethyllanosterol	−0.02	−0.07	−0.05
20-alpha,22-beta-Dihydroxycholesterol	15.11 *	58.84 *	1.96
22(R)-Hydroxycholesterol	−2.76	−2.15	0.61
24-epi-Campesterol	−2.68	−2.93	−0.25
24-Methylenecholesterol	−2.75	−3.16	−0.41
Calcidiol	−0.10	−1.64	−1.54
Calcitetrol	−2.91	−2.45	0.46
Calcitriol	−0.10	−0.39	−0.29
Campesterol	−0.50	−0.41	0.10
Cholesterol	−0.68	−0.84	−0.15
Episterol	0.27	0.03	−0.24
Lanosterol	12.39 *	10.00 *	−0.31
Lathosterol	−0.69	−1.38	−0.69
Pregnenolone	−1.76	−2.00	−0.24
Secalciferol	1.14	0.45	−0.69
Squalene	0.45	0.27	−0.18
S-squalene 2,3-epoxide	53.84 *	30.82 *	−0.80

* Ratio of FT/NC and VT/NC.

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
