# Peer review of "Effect of Embryo Vitrification on the Steroid Biosynthesis of Liver Tissue in Rabbit Offspring"

_ijms, 2020, doi:10.3390/ijms21228642_

Round 1

Reviewer 1 Report

That is a really interesting manuscript that reports the effect of embryo vitrification on the steroid byosynthesis of liver tissue in rabbit offspring. At general, manuscript is really well designed and written. Only rally minor revision is necessary before publication.

  1. Material and Methods:
  • Authors should state what NC means in the text. It is only stated at the Figure 7. 
  • Please reference artificial insemination technique;
  • Please, inform the temperature of the vitrification solution; also, inform the warming temperature.
  • Please, briefly inform the synchronization protocol used for foster mothers;

2. Results

  • Authors should inform the average efficiency of embryo transfer, showing data related to number of embryos tranferred and births. Authors should also state the number of fetuses or newborns used for each group. I was only able to realize this information at the figure tittle, in which they inform that 40 individuals per group were used to measure CR lenght.
  • I am curious regarding the mortality rate in embryos and fetuses previously submited or not to embryo vitrification. Do you have this information? If yes, please provide it.
  • Males are generally heavier than females. Did you distribute males and females equaly among experimental groups? Could you consider this variable at the discussion?

Author Response

Here, a step-by-step response is offered to each of the points indicated by the reviewer, addressing those modifications to it may contribute to the final publication of the manuscript. 

That is a really interesting manuscript that reports the effect of embryo vitrification on the steroid biosynthesis of liver tissue in rabbit offspring. At general, the manuscript is really well designed and written. Only minor rally revision is necessary before publication.

Material and Methods:

Authors should state what NC means in the text. It is only stated at Figure 7.

This suggestion has been implemented (Line 292-296).

Please reference artificial insemination technique;

This information has been implemented in the manuscript, including a reference [62]. (Line 318)

[62] Marco-Jiménez, F.; Lavara, R.; Jiménez-Trigos, E.; Vicente, J.S. In vivo development of vitrified rabbit embryos: Effects of vitrification device, recipient genotype, and asynchrony. Theriogenology 2013, 79.

Please, inform the temperature of the vitrification solution; also, inform the warming temperature.

During vitrification solutions were maintained at room temperature (around 22ºC).  Warming was performed by immersion of the straw in a water bath at 20 °C. This information has been implemented in the manuscript (Line 330-331).

Please, briefly inform the synchronization protocol used for foster mothers;

Only receptive females (determined by vulva colour) were induced to ovulate by injection of 1 μg of buserelin acetate (Hoechst Marion Roussel S.A., Madrid, Spain) 60 hours (asynchrony or −12 hours) before transfer [62]. (Line 345-347).

[62] Marco-Jiménez, F.; Lavara, R.; Jiménez-Trigos, E.; Vicente, J.S. In vivo development of vitrified rabbit embryos: Effects of vitrification device, recipient genotype, and asynchrony. Theriogenology 2013, 79.

2.Results

Authors should inform the average efficiency of embryo transfer, showing data related to the number of embryos tranferred and births. Authors should also state the number of fetuses or newborns used for each group. I was only able to realize this information at the figure tittle, in which they inform that 40 individuals per group were used to measure CR lenght.

This information has been implemented in the manuscript (Line 84-87) and also in the Fig 7.

I am curious regarding the mortality rate in embryos and fetuses previously submited or not to embryo vitrification. Do you have this information? If yes, please provide it.

These results have been widely described in our previous studies (see reference list). In order not to disturb attention from this study, survival data were not included in the earlier version. However, based on the reviewer comment, we have included general results showing overall competence. 

References:

Saenz-de-Juano MD, Marco-Jiménez F, Peñaranda DS, Joly T, Vicente JS. Effects of slow freezing procedure on late blastocyst gene expression and survival rate in rabbit. Biol Reprod. 2012:18;87:91. doi: 10.1095/biolreprod.112.100677.

J.S. Vicente, M.D. Saenz-de-Juano, E. Jiménez-Trigos, M.P. Viudes-de-Castro, D.S. Peñaranda, F. Marco-Jiménez. Rabbit morula vitrification reduces early foetal growth and increases losses throughout gestation. Cryobiology. 2013;67:321-326.

M.D Saenz-de-Juano, F. Marco-Jimenez, B. Schmaltz-Panneau, E. Jimenez-Trigos, M.P Viudes-de-Castro, D.S. Peñaranda, L. Jouneau, J. Lecardonnel, R. Lavara, C. Naturil-Alfonso, V.Duranthon, J.S Vicente. Vitrification alters at transcriptomic and proteomic level rabbit foetal placenta. Reproduction. 2014;147: 789-801.

M.D Saenz-de-Juano, F. Marco-Jimenez, M.P Viudes-de-Castro, R. Lavara, J.S Vicente. Direct comparison of the effects of slow freezing and vitrification on late blastocyst gene expression, development, implantation and offspring of rabbit morulae. Reproduction in Domestic Animals. 2014; 49: 505-511.

Saenz-de-Juano, F. Marco-Jiménez, K. Hollung, J.S. Vicente. Effects of vitrification on rabbit foetal placenta proteome at two different times of gestation. PLoS One. 2015;10:e0125157. 

Saenz-de-Juano M.D, Francisco Marco-Jiménez, Jose Salvador Vicente. Embryo transfer manipulation cause gene expression variation in blastocysts that disrupt implantation and offspring rates at birth in rabbit. European Journal of Obstetrics & Gynecology and Reproductive Biology. 2016;207:50-55.

Ximo Garcia-Dominguez, Jose S Vicente, Francisco Marco-Jiménez, Developmental plasticity in response to embryo cryopreservation: the importance of vitrification device. Animals (Basel). 2020;10:804; doi:10.3390/ani10050804.

Ximo Garcia-Dominguez, Gianfranco Diretto, Sarah Frusciante, José Salvador Vicente, Francisco Marco-Jiménez. Metabolomic analysis reveals changes in preimplantation embryos following fresh or vitrified transfer. International Journal of Molecular Science. 21(19):E7116. doi: 10.3390/ijms21197116.

Males are generally heavier than females. Did you distribute males and females equally among experimental groups? Could you consider this variable at the discussion?

This sexual effect does not occur in the rabbit in early life, presumably because the rabbit is a politocus specie. It is well known that in the rabbit, sexual dimorphism appears after they reach puberty at approximately 13-14 weeks of age [60]. In fact, rabbit carcasses are not affected by sex at the commercial weight (9 weeks).

Our group have demonstrated mainly that no interaction between the experimental group and sex has been found for body weight from birth until adulthood (20 weeks) [11,27,59]. All of these references have been included in a revised paragraph (Line 303-304).

References:

[11] Garcia-Dominguez, X.; Vicente, J.S.; Marco-Jiménez, F. Developmental plasticity in response to embryo cryopreservation: The importance of the vitrification device in rabbits. Animals 2020, 10.

[27] Garcia-Dominguez, X.; Marco-Jiménez, F.; Peñaranda, D.S.; Vicente, J.S. Long-Term Phenotypic and Proteomic Changes Following Vitrified Embryo Transfer in the Rabbit Model. Animals 2020, 10, 1043.

[59] Garcia-Dominguez, X.; Juarez, J.D.; Vicente, J.S.; Marco-Jiménez, F. Impact of embryo technologies on secondary sex ratio in rabbit. Cryobiology 2020.

[60] Ouhayoun, J. Croissance et qualités bouchères du lapin. Cuniculture 1984, 11, 181–188.

Reviewer 2 Report

Regarding manuscript # ijms-961516 entitled “Effect of embryo vitrification on the steroid biosynthesis of liver tissue in rabbit offspring”, the author aimed to unravel the particular effects related to stress induced by embryo transfer and vitrification techniques on offspring phenotype from the foetal period through to prepuberal age, using a rabbit model. In addition, the focus was extended to the liver function at prepuberal age.

The main defect in this research is absence of the data related to rabbit parents. The weight of rabbit mothers (donner and recipient) can affect weight of the offspring. Additionally, the weight of rabbit father can affect weight of offspring even with using of pooled semen. The nutrition and feeding during gestation length and after delivery until weaning can affect the weight of the offspring. The authors must fix these previous factors by using males with similar body weight, mothers with similar body weight, same feed, feeding frequency and amount and keeping these animals under same environmental condition.

Line 299 “Taking into account this fact, no sex effect was expected.” Why? Where is the reference? Usually male offspring has high weight than female at birth and weaning. The authors must include the effect of sexing their statistical model to be sure if there is effect for sex or not.

How many embryos were transferred per one mother rabbit in FT and VT groups?

How many offspring were born per one mother rabbit in all groups? The number of offspring per mother can affect weight offspring at delivery and may be at weaning.

Was there any newborn suffered from any disease like diarrhea or respiratory manifestation during the period after delivery, after weaning, prepubertal age?

Foetal and postnatal growth performance, body weight and organ phenotype were affected by all previous factors. Please add all previous requested data in the revised manuscript.

“Hepatic functionality assessment” why the authors made hepatic functionality assessment only one time at nine weeks old “prepubertal age”? why they think that embryo vitrification can affect hepatic functionality assessment only one time at nine weeks old “prepubertal age” and not at delivery or weaning age?!

In table 1, please add (FT/VT) “Targeted identification of differentially accumulated metabolites in liver tissue of prepuberal offspring born after embryo manipulation during the transfer procedure compared to vitrified embryos procedure in steroid biosynthesis pathway.”

Author Response

Here, a step-by-step response is offered to each of the points indicated by the reviewer, addressing those modifications to it may contribute to the final publication of the manuscript. 

Regarding manuscript # ijms-961516 entitled “Effect of embryo vitrification on the steroid biosynthesis of liver tissue in rabbit offspring”, the author aimed to unravel the particular effects related to stress induced by embryo transfer and vitrification techniques on offspring phenotype from the foetal period through to prepuberal age, using a rabbit model. In addition, the focus was extended to the liver function at prepuberal age.

The main defect in this research is absence of the data related to rabbit parents. The weight of rabbit mothers (donner and recipient) can affect weight of the offspring. Additionally, the weight of rabbit father can affect weight of offspring even with using of pooled semen. The nutrition and feeding during gestation length and after delivery until weaning can affect the weight of the offspring. The authors must fix these previous factors by using males with similar body weight, mothers with similar body weight, same feed, feeding frequency and amount and keeping these animals under same environmental condition.

We completely understand the remarks of the reviewer attending the nonexistence parents information. All these subjects were controlled to try to diminish as much as possible the bias effects. All parents used in the experiment (donors, recipients, and males) were 20 weeks old. All of them were reared together under the same environmental conditions (same facility room), including feed, and the same diet regimen. All animals were housed individually at 12 weeks of age, with free access to fed and water, under a 16-hour light/8-hour dark photoperiod. At 16 weeks of age, animals were fed with 130 g/day to comply with energy requirements for maintenance (340 kJ day-1 kg-1 LW0.75; Xiccato and Trocino 2010). Under this feeding regimen, the average body weight of the 10 males used to obtain the pooled semen to do the artificial insemination was 3675.0±141.7 g. The average body weight of the 20 naturally conceived females was 3635.3±307.3 g (8 from control group, NC- and 12 donor does to produce embryos to transfer group-FT- and vitrified group-VT-). The average body weight of the 34 foster mothers was 3736.0± 305.6 g and 3668.4± 336.7 g for fresh embryos and vitrified embryos, respectively. Finally, the 10 naturally conceived females that gave birth present an average bodyweight of 3718.0±321.7 g. Under these conditions, the embryos are remarkably randomized. Moreover, and given the considerable number of animals produced, we assume that the bias effect has been diminished to the maximum, giving power to the results obtained. All this data has been included in the manuscript.

Xiccato,G Trocino A. 2010. Energy and Protein Metabolism and Requirements. In: de Blas C, Wiseman J (ed), Nutrition of the Rabbit 2nd edition. CABI, Wallingford, UK, pp. 83-118.

Line 299 “Taking into account this fact, no sex effect was expected.” Why? Where is the reference? Usually male offspring has high weight than female at birth and weaning. The authors must include the effect of sexing their statistical model to be sure if there is effect for sex or not.

This sexual effect does not occur in the rabbit in early life, presumably because the rabbit is a politocus specie. It is well known that in the rabbit, sexual dimorphism appears after they reach puberty at approximately 13-14 weeks of age [60]. In fact, rabbit carcasses are not affected by sex at the commercial weight (9 weeks).

Our group have demonstrated mainly that no interaction between the experimental group and sex has been found for body weight from birth until adulthood (20 weeks) [11,27,59]. All of these references have been included in a revised paragraph (Line 303-304).

References:

[11] Garcia-Dominguez, X.; Vicente, J.S.; Marco-Jiménez, F. Developmental plasticity in response to embryo cryopreservation: The importance of the vitrification device in rabbits. Animals 2020, 10.

[27] Garcia-Dominguez, X.; Marco-Jiménez, F.; Peñaranda, D.S.; Vicente, J.S. Long-Term Phenotypic and Proteomic Changes Following Vitrified Embryo Transfer in the Rabbit Model. Animals 2020, 10, 1043.

[59] Garcia-Dominguez, X.; Juarez, J.D.; Vicente, J.S.; Marco-Jiménez, F. Impact of embryo technologies on secondary sex ratio in rabbit. Cryobiology 2020.

[60] Ouhayoun, J. Croissance et qualités bouchères du lapin. Cuniculture 1984, 11, 181–188.

How many embryos were transferred per one mother rabbit in FT and VT groups?

This was previously included in the previous version. In both the FT and VT groups, between 12-14 were transferred to each female. Half (6-7) were transferred in each of the uterine horns (independent uteri since the rabbit is bicervical). (Line 323)

How many offspring were born per one mother rabbit in all groups? The number of offspring per mother can affect weight offspring at delivery and maybe at weaning.

Litter size was 9.7±0.37 for NC group, 7.7±0.38 for FT group and 7.1±0.45 for VT. In the statistical section, it was indicated that litter size was used as a covariate for data correction since birth until prepuberal age.  Litter size and total offspring data have been implemented in the manuscript (Line 84-87).

Was there any newborn suffered from any disease like diarrhea or respiratory manifestation during the period after delivery, after weaning, prepubertal age?

The study is based solely on animals without any apparent pathology. Mortality from birth to adulthood was similar between all experimental groups, range within common value in the rabbit.

Foetal and postnatal growth performance, body weight and organ phenotype were affected by all previous factors. Please add all previous requested data in the revised manuscript.

Answered above

“Hepatic functionality assessment” why the authors made hepatic functionality assessment only one time at nine weeks old “prepubertal age”? why they think that embryo vitrification can affect hepatic functionality assessment only one time at nine weeks old “prepubertal age” and not at delivery or weaning age?

This study is based on several studies that our group have been developing for some years. In them, we have regularly observed that the manipulation of early embryos induces changes in growth/body weight and liver weight, related with liver function (see references). In this study, we suggest that long term phenotypic variation and the changes in hepatic function observed after embryo manipulation may be associated, to great extent, with the embryo transfer.

  1. Ximo Garcia-Dominguez, Jose S. Vicente, Francisco Marco-Jimenez. Does the embryo vitrification procedure impact the secondary sex ratio? CRYOBIOLOGY. S0011-2240(20)30292-3. doi: 10.1016/j.cryobiol.2020.10.008.
  2. Ximo Garcia-Dominguez, Gianfranco Diretto, Sarah Frusciante, José Salvador Vicente, Francisco Marco-Jiménez. Metabolomic analysis reveals changes in preimplantation embryos following fresh or vitrified transfer. International Journal of Molecular Science. 21(19):E7116. doi: 10.3390/ijms21197116.
  3. Ximo Garcia-Dominguez, José Salvador Vicente, María Pilar Viudes-de-Castro, Francisco Marco-Jimenez. Long-term effects following fresh/vitrified embryo transfer are transmitted by paternal germline in rabbits. Animals (Basel) 2020;10;1272. doi:10.3390/ani10081272s.
  4. Ximo Garcia-Dominguez, Francisco Marco-Jimenez, David S Peñaranda, Victor Garcia-Carpintero, Joaquín Cañizares, Jose S Vicente. Long-term and transgenerational phenotypic, transcriptional and metabolic effects in rabbit males born following vitrified embryo transfer. Scientific Reports 2020;10:11313. doi: 10.1038/s41598-020-68195-9.
  5. Ximo Garcia-Dominguez, Francisco Marco-Jimenez, David S Peñaranda, Jose S Vicente. Embryo cryopreservation-transfer procedure entails long-term phenotypic consequences correlated with molecular signatures in the rabbit model. Animals (Basel). 2020;10:1043; doi:10.3390/ani10061043.
  6. Ximo Garcia-Dominguez, Jose S Vicente, Francisco Marco-Jiménez, Developmental plasticity in response to embryo cryopreservation: the importance of vitrification device. Animals (Basel). 2020;10:804; doi:10.3390/ani10050804.
  7. Saenz-de-Juano M.D, Francisco Marco-Jiménez, Jose Salvador Vicente. Embryo transfer manipulation cause gene expression variation in blastocysts that disrupt implantation and offspring rates at birth in rabbit. European Journal of Obstetrics & Gynecology and Reproductive Biology. 2016;207:50-55.
  8. Lavara R, Baselga M, Marco-Jiménez F, Vicente JS. Embryo vitrification in rabbits: consequences for progeny growth. Theriogenology 2015;84:674-80.
  9. Saenz-de-Juano, F. Marco-Jiménez, K. Hollung, J.S. Vicente. Effects of vitrification on rabbit foetal placenta proteome at two different times of gestation. PLoS One. 2015;10:e0125157.
  10. D Saenz-de-Juano, F. Marco-Jimenez, M.P Viudes-de-Castro, R. Lavara, J.S Vicente. Direct comparison of the effects of slow freezing and vitrification on late blastocyst gene expression, development, implantation and offspring of rabbit morulae. Reproduction in Domestic Animals. 2014; 49: 505-511.
  11. D Saenz-de-Juano, F. Marco-Jimenez, B. Schmaltz-Panneau, E. Jimenez-Trigos, M.P Viudes-de-Castro, D.S. Peñaranda, L. Jouneau, J. Lecardonnel, R. Lavara, C. Naturil-Alfonso, V.Duranthon, J.S Vicente. Vitrification alters at transcriptomic and proteomic level rabbit foetal placenta. Reproduction. 2014;147: 789-801.
  12. Lavara R, Baselga M, Marco-Jiménez F, Vicente J.S. Long-term and transgenerational effects of cryopreservation on rabbit embryos. Theriogenology. 2014:81;988-992.
  13. Saenz-de-Juano MD, Marco-Jiménez F, Peñaranda DS, Joly T, Vicente JS. Effects of slow freezing procedure on late blastocyst gene expression and survival rate in rabbit. Biol Reprod. 2012:18;87:91. doi: 10.1095/biolreprod.112.100677.

In table 1, please add (FT/VT) “Targeted identification of differentially accumulated metabolites in liver tissue of prepuberal offspring born after embryo manipulation during the transfer procedure compared to vitrified embryos procedure in steroid biosynthesis pathway.”

It has been included

Round 2

Reviewer 2 Report

Thanks for your kind response. The manuscript was improved and can be accepted in its present form.

Author Response

Thank you very much for your kind words and support for the publication of this study.